# MVGE: Scale-invariant and Temporal-consistent Monocular Video Geometry Estimation

## Abstract

We present MVGE, a novel approach for estimating 3D geometry from extended monocular video sequences, where existing methods struggle to maintain both geometric accuracy and temporal consistency across hundreds of frames. Our approach generates affine-invariant 3D point maps with shared parameters across entire sequences, enabling consistent scale-invariant representations. We introduce three key innovations: viewpoint-invariant geometry aligning multi-perspective points in a unified reference frame; appearance-invariant learning enforcing consistency across exponential timescales; and frequency-modulated positioning enabling extrapolation to sequences vastly exceeding training length. Experiments across diverse datasets demonstrate significant improvements, reducing relative point map error by 24.2% and temporal alignment error by 34.9% on ScanNet compared to state-of-the-art methods. Our approach handles challenging scenarios with complex camera trajectories and lighting variations while efficiently processing extended sequences in a single pass. Code will be publicly released, and we encourage readers to explore the interactive demonstrations in our supplementary materials.

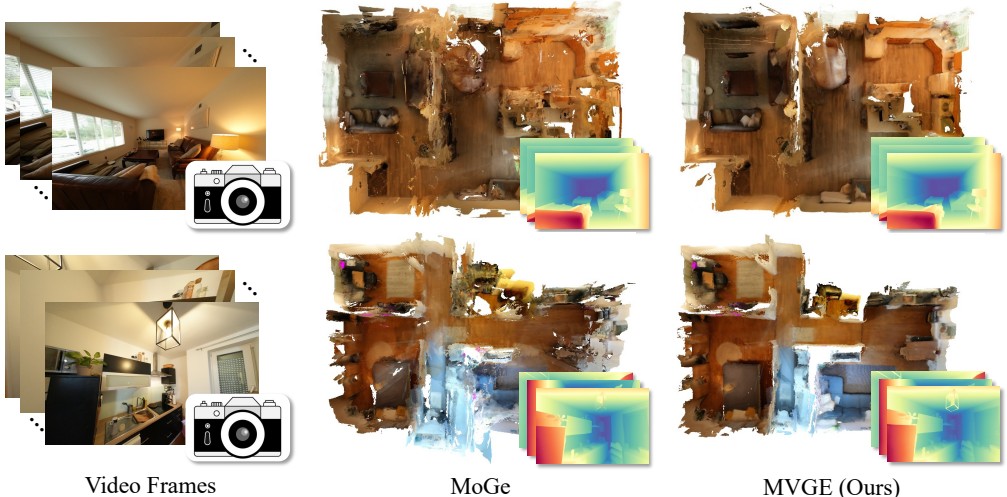

| Video Frames | MoGe | MVGE (Ours) |

Figure 1: Given a sequence of video frames, MVGE is capable of predicting scale-invariant and temporal-consistent point maps in a single forward pass. We visualize the 3D mesh reconstructed by TSDF integration of 100 point maps predicted by MVGE in a single shot, in comparison with MoGe (Wang et al., 2025b) using ScanNet++ (Yeshwanth et al., 2023) dataset. MVGE maintains geometric accuracy and long-range consistency across hundreds of frames with minimum drift, enabling high-quality 3D reconstruction.

# 1 INTRODUCTION

Estimating 3D geometry from monocular videos is a fundamental challenge in computer vision with diverse applications in novel view synthesis, autonomous navigation, virtual reality, and 3D/4D reconstruction. Despite significant advances in single-image depth estimation, video-based approaches still struggle with two critical challenges: achieving **high geometric accuracy at multiple scales** within each frame and across the global coordinate system, while maintaining **temporal consistency throughout sequences of hundreds of frames without scale drift**.

Existing methods typically excel in one area at the expense of the other. Single-image approaches like MoGe (Wang et al., 2025b) capture detailed geometry but produce inconsistent results when applied frame-by-frame to videos. Conversely, video-specific methods (Yang et al., 2025; Hu et al., 2025; Chen et al., 2025; Wang et al., 2024a; Zhang et al., 2025) inherently lack geometric precision while providing only short-term consistency, still exhibiting significant scale drift in longer sequences. Traditional approaches rely on optical flow constraints (Wang et al., 2023; 2024b) that only link adjacent frames, failing to prevent accumulation of errors. Video diffusion models offer consistency through learned priors but at significant computational cost. Recent transformer-based approaches like VGGT (Wang et al., 2025a) can process longer sequences but lack effective temporal position encoding, limiting their effectiveness with complex camera motions.

Processing lengthy sequences with **geometric and temporal accuracy** requires **simultaneous consideration of hundreds of frames** with **precise temporal position encoding** to handle complex scene transformations. However, memory constraints make training on such long sequences impractical. This creates a fundamental tension: models need to **extrapolate effectively** to sequences far longer than their training examples. With robust extrapolation capabilities, overlapping inference techniques can achieve minimal drift by maintaining substantial frame overlap between consecutive windows, effectively scaling to unlimited sequence lengths.

We present a novel approach generating **affine-invariant** 3D point maps from RGB videos with both geometric precision and long-range temporal consistency. Our method produces point maps where all frames share the same scale and shift parameters, with a unified optimization approach to recover scale-invariant representations for downstream applications. Our key innovations include: **Viewpoint-invariant geometry** transforming points from multiple perspectives into a shared reference frame through camera pose integration; **Appearance-invariant learning** that supervises geometric consistency across exponential time scales while isolating persistent structural features from transient visual conditions; and **Adaptive frequency-modulated positioning** implementing an NTK-guided rotary scheme with strategic training-time extrapolation simulation to process sequences orders of magnitude longer than training examples. Our approach significantly outperforms previous methods, reducing relative point map error by 24.2% on ScanNet and temporal alignment error by 34.9% compared to existing approaches, while maintaining superior performance across diverse datasets from synthetic animations to real-world driving scenarios.

# 2 RELATED WORK

**Monocular depth estimation.** Recent advances in monocular depth estimation have significantly improved both geometric accuracy and generalization. Early supervised approaches (Eigen et al., 2014; Fu et al., 2018; Bhat et al., 2021; 2023) were limited by domain-specific datasets. More recent methods overcame this limitation through affine-invariant representations (Ranftl et al., 2022; Birkl et al., 2023; Ranftl et al., 2021) or scale alignment techniques (Yin et al., 2023; Hu et al., 2024). Large-scale data-driven approaches (Yang et al., 2024a;b) and diffusion-based models (Ke et al., 2024; Gui et al., 2024; Fu et al., 2024) have further enhanced generalization to diverse scenarios. While some methods (Yin et al., 2021b; Piccinelli et al., 2024; 2025; Bochkovskii et al., 2025) predict both depth and camera intrinsics, they often lack precision in local geometry. MoGe (Wang et al., 2025b) achieves superior geometric accuracy through multi-scale supervision but operates only on single images, lacking cross-frame consistency.

**Video-based depth estimation.** Extending depth estimation to video sequences introduces significant temporal consistency challenges. Video diffusion models (Hu et al., 2025; Yang et al., 2025)

provide inherent coherence but at high computational cost. For sequences longer than training examples, several strategies have emerged: sliding windows (Hu et al., 2025; Chen et al., 2025), keyframe conditioning (Yang et al., 2025), and global attention (Wang et al., 2025a). However, these methods still exhibit scale drift over extended sequences or struggle with complex camera trajectories. Current approaches typically excel at either geometric accuracy or temporal consistency, rarely achieving both across hundreds of frames.

**Positional encoding for extrapolation.** Transformers struggle with sequences longer than their training examples. While standard sinusoidal encodings (Vaswani et al., 2017) and learned embeddings have limited extrapolation capabilities, Rotary Position Encoding (RoPE) (Su et al., 2021) better generalizes by encoding relative positions through complex plane rotations. Strategic frequency adjustments (Chen et al., 2023; Peng et al., 2024) and NTK-aware adaptations (Peng & Quesnelle, 2023; Sun et al., 2022) preserve both local details and global structure during extrapolation. Our work adapts these techniques, primarily developed for language models, to video-based 3D reconstruction, enabling effective processing of sequences substantially longer than training examples.

## 3 METHOD

We present a novel approach for generating geometrically accurate and temporally consistent 3D point maps from RGB videos. Our method addresses two critical challenges: producing geometrically precise representations for each frame, and maintaining long-range temporal consistency across hundreds of frames - essential requirements for downstream 3D reconstruction tasks.

### 3.1 GEOMETRY-AWARE VIDEO POINT MAP ESTIMATION

**Task definition.** Given an RGB video sequence $\mathbf{I} = \{I_1, I_2, ..., I_T\}$ with $T$ frames, our goal is to estimate scale-invariant and temporally consistent 3D point maps from unposed monocular videos. Specifically, we predict a sequence of 3D point maps $\mathbf{P} = \{P_1, P_2, ..., P_T\}$, where $P_t \in \mathbb{R}^{H \times W \times 3}$ represents the 3D coordinates of each pixel in frame $t$ within that frame's camera coordinate system. **Training setup:** During training, our method takes multi-frame RGB images as network input and optionally uses ground truth camera poses solely for computing the cross-frame geometric loss. The poses enable multi-scale geometric supervision by transforming predicted point clouds to a common reference frame, but are not required for all training data. **Inference setup:** At inference, our method requires only multi-frame RGB images as input and outputs scale-consistent point maps in each frame's camera coordinate system. These point maps can subsequently serve as input to methods like MegaSAM (Li et al., 2025) to estimate camera parameters and enable high-quality 4D reconstruction.

**Positioning relative to existing approaches.** Our method addresses fundamental limitations of existing approaches across three categories: **Single-frame pointmap methods** (e.g., Depth-Pro (Bochkovskii et al., 2025), MoGe (Wang et al., 2025b)) process frames independently, leading to scale inconsistencies that degrade downstream reconstruction quality. Our approach achieves superior long-range temporal consistency and global geometric accuracy, enabling more precise camera pose estimation and 4D reconstruction. **Video depth methods** (e.g., DepthCrafter (Hu et al., 2025), Video Depth Anything (Chen et al., 2025)) typically output affine-invariant depth maps, where the missing shift parameter and camera intrinsics complicate direct 4D reconstruction. Compared to these video depth prediction methods, our approach maintains scale consistency across significantly longer temporal sequences. **Single coordinate system methods** (e.g., Dust3r (Wang et al., 2024a), MonST3R (Zhang et al., 2025)) directly estimate global point maps and camera poses jointly. While our approach requires external pose estimation, it produces more accurate 4D reconstructions and handles significantly longer video sequences under identical memory constraints.

**Scale-Invariant representation.** Our model predicts affine-invariant point maps following MoGe (Wang et al., 2025b), where each point map is agnostic to global scale $s \in \mathbb{R}$ and offset $\mathbf{t} \in \mathbb{R}^3$. The key distinction is that our entire video sequence shares these parameters: $P_i \cong sP_i + \mathbf{t}, \forall i \in [1, T]$. During inference, we recover a single shared focal length $f$ and Z-axis shift $t_z$ for all frames by minimizing the projection error:

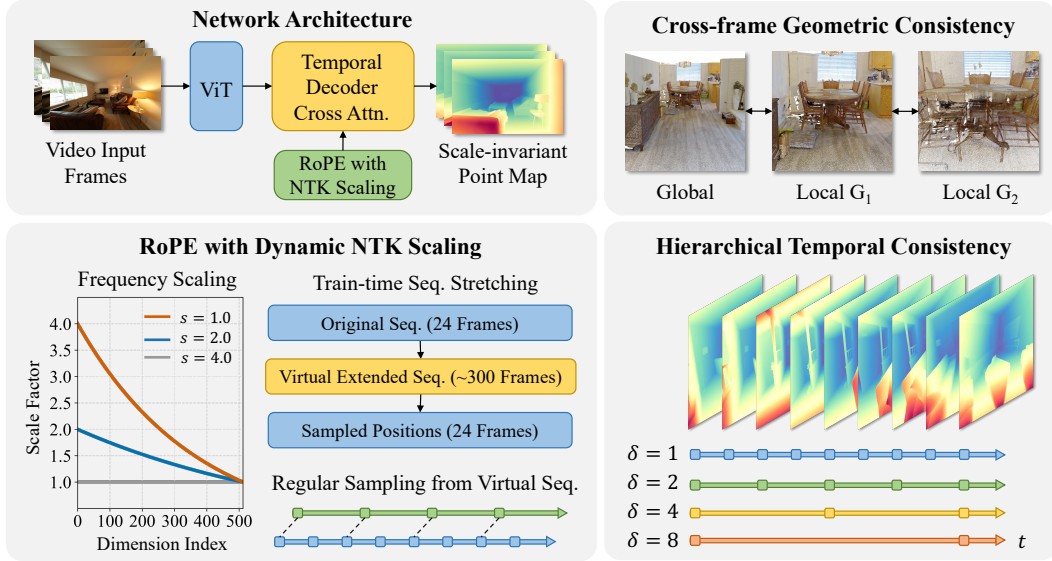

Figure 2: **Overview of MVGE.** *Top-Left:* MVGE consists of a ViT backbone that processes video input frames, followed by a temporal decoder with cross-attention and dynamic NTK scaling RoPE, producing scale-invariant point maps (Sec. 3.1). *Top-Right:* Cross-frame geometric consistency enforced across global and local geometric levels ($G_1$, $G_2$) to maintain structural coherence across frames (Sec. 3.1). *Bottom-Left:* RoPE with dynamic NTK scaling applied to extend sequence context, using frequency scaling that adaptively weights dimensions based on scale factor, and train-time sequence stretching that creates a virtual extended sequence to sample positions (Sec. 3.2). *Bottom-Right:* Hierarchical temporal consistency constraints applied multiple temporal strides ($\delta = 1, 2, 4, 8$) to enforce smooth, consistent point map predictions across time (Sec. 3.2).

$$\min_{f,t_z} \sum_{t=1}^{T} \sum_{i=1}^{N} \left( \frac{fx_{t,i}}{z_{t,i} + t_z} - u_{t,i} \right)^2 + \left( \frac{fy_{t,i}}{z_{t,i} + t_z} - v_{t,i} \right)^2, \tag{1}$$

where $(x_{t,i}, y_{t,i}, z_{t,i})$ are the predicted 3D coordinates and $(u_{t,i}, v_{t,i})$ are the corresponding 2D pixel coordinates. This ensures a metrically consistent representation across the entire video, essential for 3D reconstruction tasks. By recovering the shift parameter, we transform our predictions into scale-invariant point maps, making them directly applicable for downstream tasks such as 3D reconstruction and novel view synthesis.

**Geometric accuracy through multi-scale training.** To achieve fine-grained geometric accuracy, we adopt MoGe's multi-scale approach (Wang et al., 2025b). MoGe achieves superior geometric precision through three key mechanisms: (1) affine-invariant alignment that handles ambiguities in depth perception, (2) multi-scale local geometry supervision that enforces accuracy at different spatial scales, and (3) normal consistency loss that ensures surface coherence. These mechanisms collectively enable the capture of both global structure and fine geometric details.

Building on this foundation, our approach extends the geometric accuracy requirements to video sequences. Additionally, other works (Bochkovskii et al., 2025; Yang et al., 2024b; Yin et al., 2021a; 2019) have demonstrated that spatial gradient regularization can further improve detailed depth estimation by constraining the local surface structure. We incorporate these principles into our video-based framework to maintain high-fidelity geometric representations across frames.

**Cross-frame geometric constraints.** To enforce geometric consistency across frames, we transform all points to a common reference frame using camera poses. We randomly select one frame from the sequence as the reference frame for each training iteration, providing diverse viewpoints during training. This process involves first converting points from individual camera coordinates to world coordinates using the camera-to-world transforms, and then transforming these world points to the

randomly selected reference frame. This transformation allows us to directly compare geometric structures captured from different viewpoints within a unified coordinate system.

We then apply a multi-scale geometric loss framework to the points in this common reference frame:

$$\mathcal{L}_{cross} = \sum_{l \in \{1, G_1, G_2\}} \frac{1}{|C_l|} \sum_{c \in C_l} \frac{1}{|M_c|} \sum_{i \in M_c} w_i \cdot \|s_c \cdot \mathbf{p}_{pred}^{ref}[i] + \mathbf{t}_c - \mathbf{p}_{gt}^{ref}[i]\|_1, \qquad (2)$$

where $l$ is the grid size (with $l = 1$ representing global alignment), $C_l$ is the set of cells at grid size $l$, $M_c$ is the set of valid points in cell $c$, $w_i$ is a depth-aware weight, and $(s_c, \mathbf{t}_c)$ are alignment parameters computed independently for each cell.

For global alignment ($l = 1$), the entire point cloud is treated as a single cell. For local alignment, we divide the 3D space into a grid of $G_l \times G_l \times G_l$ cells. Our implementation uses grid sizes of 4 and 16, allowing the model to capture both coarse structure and fine details across the entire temporal sequence. By enforcing geometric consistency at multiple scales, our approach ensures that the predicted point maps maintain both local detail and global structure across the video.

## 3.2 LONG-RANGE TEMPORAL CONSISTENCY

**Temporal consistency challenges.** Downstream reconstruction tasks require point maps that exhibit: (1) consistent geometric accuracy both within individual frames and across the entire sequence at local and global scales, and (2) temporal stability over extended sequences rather than just between adjacent frames. When these requirements aren't met, particularly under challenging conditions with dramatic lighting changes or significant camera movements, scale drift can accumulate, severely degrading reconstruction quality and producing distorted or fragmented results.

Recent video diffusion model-based approaches (Hu et al., 2025; Yang et al., 2025; Shao et al., 2025) leverage inherent temporal consistency mechanisms, but suffer from significant computational inefficiency. Other methods utilize optical flow-based losses (Wang et al., 2023; 2024b; Kuang et al., 2025; Chen et al., 2025) to maintain consistency between adjacent frames. However, these approaches only constrain relationships between consecutive frames, causing error accumulation over longer sequences. Furthermore, they struggle with large camera motions, which can substantially degrade depth prediction accuracy by introducing conflicting geometric constraints when camera viewpoint changes significantly.

**Structure-Preserving temporal supervision.** To address fundamental limitations in temporal consistency, we introduce a hierarchical derivative supervision framework that operates across multiple time scales:

$$\mathcal{L}_{temp} = \sum_{s=0}^{S-1} \frac{1}{|M_s|} \sum_{t=1}^{T-\delta_s} \sum_{i \in \mathcal{M}_{t, t+\delta_s}} w_{t,i} \cdot \left| \frac{\partial D_{pred}}{\partial t}(t, i) - \frac{\partial D_{gt}}{\partial t}(t, i) \right|, \qquad (3)$$

where $s$ indexes temporal scale, $S$ is the total number of scales, $\delta_s = 2^s$ represents exponentially increasing time intervals, $T$ is the sequence length, $\mathcal{M}_{t, t+\delta_s}$ denotes valid corresponding pixels between frames $t$ and $t + \delta_s$, $|M_s|$ is the total number of valid pixels at scale $s$, $w_{t,i}$ is a depth-aware weight for pixel $i$ in frame $t$, and $\frac{\partial D}{\partial t}$ represents the temporal derivative of depth values.

To disentangle geometric structure from appearance variations, we apply frame-specific augmentations with independently sampled color transformations and blur patterns across the sequence. This forces the model to focus on invariant geometric features while ignoring transient visual cues, enabling robust geometric consistency even under dramatic lighting changes and complex camera movements that typically challenge conventional methods.

**Scaling beyond memory constraints.** Processing hundreds of frames simultaneously during training is infeasible due to memory constraints. Existing methods address this limitation in various ways: DepthCrafter (Hu et al., 2025) and Video Depth Anything (Chen et al., 2025) use overlapping frame windows during inference, but suffer from scale drift due to limited training sequence length. Depth Anything Video (Yang et al., 2025) processes key frames first and then uses them as conditions for

other frames, but this approach has limited scalability and reduced efficiency. VGGT (Wang et al., 2025a) employs global attention without temporal information injection, struggling with complex camera trajectories where temporal relationships are critical.

**Frequency-modulated extrapolation.** To ensure robust handling of complex spatial relationships while enabling effective extrapolation to sequences much longer than those seen during training, we employ a specialized Rotary Position Encoding (RoPE) (Su et al., 2021; Chen et al., 2023; Peng & Quesnelle, 2023; Sun et al., 2022) with Neural Tangent Kernel (NTK) adaptation.

Our implementation computes frequency components with dynamic NTK scaling:

$$\theta_{i,j} = \frac{j \cdot s^{(1-\frac{i}{d})}}{10000^{\frac{2i}{d}}}, \tag{4}$$

where $\theta_{i,j}$ is the rotation angle, $j$ is the position index, $i$ indexes the frequency dimension, $d$ is the embedding dimension, and $s = \frac{L_{seq}}{L_{train}}$ is a scaling factor applied when inference sequence length exceeds training length. This adaptive scaling preserves the model's capacity to capture both fine-grained temporal patterns and global structure by applying graduated adjustments across the frequency spectrum—attenuating changes to high-frequency components that encode local details while amplifying adjustments to low-frequency components that capture long-range dependencies.

During training, we randomly apply sequence stretching with 50% probability, where we generate position encodings for a virtual extended sequence and sample them at appropriate intervals to match the original sequence length. Mathematically, this involves computing $\theta'_{i,j}$ for a virtual sequence of length $L_{virtual} = L_{seq} \cdot r$ (where $r$ is randomly sampled) and then sampling positions $j' = j \cdot r$ to obtain the final encodings. This technique simulates extrapolation during training, teaching the model to handle sequences significantly longer than those in the training data.

To further enhance temporal generalization, we employ variable temporal context windows during training. While maintaining a fixed 24-frame input size, we dynamically adjust the temporal stride between frames, allowing these 24 frames to represent contexts spanning from densely sampled short sequences to sparsely sampled long sequences of several hundred frames. This adaptive sampling strategy complements our position encoding approach, enabling the model to simultaneously learn representations for both fine-grained frame-to-frame transitions and long-range temporal relationships.

## 4 EXPERIMENT

### 4.1 IMPLEMENTATION DETAILS

**Model architecture.** Our model builds upon MoGe (Wang et al., 2025b) by integrating temporal modeling capabilities through strategically placed temporal attention modules in the decoder. Specifically, we insert four transformer-based temporal modules after each feature level of the decoder with 8 attention heads and single-block architecture, enabling effective information exchange across frames while preserving spatial details. We employ DINOv2-L (Oquab et al., 2023) as our visual encoder and initialize all parameters from MoGe's pretrained weights.

**Training datasets.** For training, we use a diverse collection of synthetic datasets including TartanAir (Wang et al., 2020), PointOdyssey (Zheng et al., 2023), SPRING (Mehl et al., 2023), VKitti2 (Cabon et al., 2020), Lightwheel (LightwheelAI & contributors, 2024), Hypersim (Roberts et al., 2021), GTAIM (Cao et al., 2020), MVSSynth (Huang et al., 2018), UnrealStereo4K (Tosi et al., 2021), GTASFM (Wang & Shen, 2019), IRS (Wang et al., 2021), and MidAir (Fonder & Droogenbroeck, 2019).

**Optimization strategy.** We optimize using AdamW following MoGe's base configuration with learning rates of $10^{-4}$ for decoder parameters and $10^{-5}$ for encoder parameters. These rates are dynamically scaled according to the square root of batch size ratio, using a reference batch size of 32 frames as baseline. Our learning schedule employs warmup, linear decay, and step decay phases with all milestone parameters proportionally adjusted based on total iteration count. Throughout training, we preserve input aspect ratios while resizing images to maintain spatial relationships in the scene.

**Loss functions.** Our loss function integrates MoGe's original components with additional spatial and temporal consistency objectives. We maintain the affine-invariant global loss (weight 1.0), multi-scale

| Method | Sintel | | ScanNet | | | Bonn | | KITTI | | Avg. |
| | Rel$^p$↓ | δ$^p$↑ | Rel$^p$↓ | δ$^p$↑ | TAE$^p$↓ | Rel$^p$↓ | δ$^p$↑ | Rel$^p$↓ | δ$^p$↑ | Rank↓ |
|---|---|---|---|---|---|---|---|---|---|---|
| DepthPro (Bochkovskii et al., 2025) | 0.400 | 0.441 | 0.132 | 0.942 | 0.095 | 0.130 | 0.975 | 0.191 | 0.810 | 3.67 |
| VGGT (Wang et al., 2025a) | 0.382 | **0.694** | 0.032 | 0.992 | 0.079 | **0.043** | **0.987** | 0.196 | 0.764 | 2.67 |
| MoGe (Wang et al., 2025b) | 0.281 | 0.627 | 0.132 | 0.896 | 0.126 | 0.086 | 0.967 | 0.101 | 0.971 | 2.25 |
| Ours | **0.257** | 0.617 | **0.100** | **0.961** | **0.082** | 0.068 | 0.979 | **0.091** | **0.976** | **1.25** |

| | Rel$^d$↓ | δ$^d$↑ | Rel$^d$↓ | δ$^d$↑ | TAE$^d$↓ | Rel$^d$↓ | δ$^d$↑ | Rel$^d$↓ | δ$^d$↑ | Rank↓ |
|---|---|---|---|---|---|---|---|---|---|---|
| DepthPro (Bochkovskii et al., 2025) | 0.363 | 0.476 | 0.089 | 0.929 | 0.065 | 0.056 | 0.973 | 0.092 | 0.912 | 3.33 |
| VGGT (Wang et al., 2025a) | 0.359 | **0.680** | 0.029 | 0.989 | 0.048 | **0.040** | **0.981** | 0.187 | 0.728 | 2.67 |
| MoGe (Wang et al., 2025b) | 0.255 | 0.603 | 0.130 | 0.852 | 0.077 | 0.081 | 0.959 | 0.087 | 0.958 | 2.50 |
| Ours | **0.216** | 0.648 | **0.081** | **0.941** | **0.049** | 0.055 | 0.971 | **0.081** | **0.965** | **1.25** |

Table 1: **Evaluation on point map estimation and depth estimation.** Results are aligned with the ground truth by optimizing a shared scale factor across the entire video. Lower values are better for Rel and TAE (↓), while higher values are better for δ (↑). The best results in each column are highlighted in **bold**. Gray values indicate methods trained on ScanNet.

| Pos. Encoding | Sintel | | | ScanNet | | | Bonn | |
| | Rel$^p$↓ | δ$^p$↑ | TAE$^p$↓ | Rel$^p$↓ | δ$^p$↑ | TAE$^p$↓ | Rel$^p$↓ | δ$^p$↑ |
|---|---|---|---|---|---|---|---|---|
| None | 0.304 | **0.503** | 0.426 | 0.163 | 0.878 | 0.089 | 0.118 | 0.958 |
| APE | 0.324 | 0.475 | 0.451 | 0.153 | 0.895 | 0.089 | 0.115 | 0.956 |
| RoPE | 0.307 | 0.491 | 0.410 | 0.140 | 0.915 | 0.092 | 0.103 | **0.964** |
| RoPE+ | **0.304** | **0.503** | 0.394 | **0.138** | 0.923 | 0.086 | **0.095** | 0.963 |

| Pos. Encoding | Sintel | | | ScanNet | | | Bonn | |
| | Rel$^d$↓ | δ$^d$↑ | TAE$^d$↓ | Rel$^d$↓ | δ$^d$↑ | TAE$^d$↓ | Rel$^d$↓ | δ$^d$↑ |
|---|---|---|---|---|---|---|---|---|
| None | 0.261 | 0.547 | 0.246 | 0.107 | 0.896 | 0.053 | 0.073 | 0.954 |
| APE | 0.279 | 0.526 | 0.252 | 0.108 | 0.889 | 0.052 | 0.076 | 0.947 |
| RoPE | 0.261 | 0.547 | 0.235 | 0.097 | 0.911 | 0.055 | **0.062** | 0.959 |
| RoPE+ | **0.253** | **0.563** | 0.222 | **0.095** | 0.919 | 0.047 | 0.064 | **0.959** |

| Method | Sintel | | Bonn | | FPS |
| | Rel$^d$↓ | δ$^d$↑ | Rel$^d$↓ | δ$^d$↑ | |
|---|---|---|---|---|---|
| DepthCrafter | 0.30 | 0.70 | 0.13 | 0.85 | 0.94 |
| Video Depth Any. | 0.30 | 0.64 | 0.07 | 0.96 | 4.47 |
| DepthAnyVideo | 0.41 | 0.66 | **0.06** | **0.97** | 6.48 |
| **MVGE (Ours)** | **0.20** | **0.73** | **0.06** | **0.97** | **39.1** |

| Method | ScanNet | | KITTI | | Time (s) |
| | Rel$^d$↓ | δ$^d$↑ | Rel$^d$↓ | δ$^d$↑ | |
|---|---|---|---|---|---|
| DepthCrafter | 0.17 | 0.73 | 0.15 | 0.77 | 320.1 |
| Video Depth Any. | **0.09** | 0.92 | 0.08 | 0.95 | 67.2 |
| DepthAnyVideo | **0.09** | **0.93** | 0.11 | 0.89 | 46.3 |
| **MVGE (Ours)** | **0.09** | **0.93** | **0.07** | **0.97** | **7.7** |

Table 2: **Ablation study on extrapolation strategies.** Position encoding methods on 270-frame sequences exceeding our 24-frame training sequences. RoPE+ combines NTK-adapted rotary encoding with sequence stretching training.

Table 3: **Video depth methods comparison.** Evaluation on 300 frames at 378×672 resolution with affine-invariant alignment.

local losses at levels 4, 16, and 64 (weights 1.0 each), normal loss (1.0), and mask loss (1.0). We adopt established spatial gradient loss (4.0) to preserve depth details, and introduce our proposed $\mathcal{L}_{temp}$ (2.0) and $\mathcal{L}_{cross}$ (1.0). For frame-specific augmentation, we apply color jitter and Gaussian blur with 0.5 probability to enhance robustness to appearance variations.

**Computational resources.** We trained our final model on 16 NVIDIA H20 GPUs for approximately 4.3 days. Each ablation study experiment required approximately 0.6 days of training on the same hardware configuration.

## 4.2 EVALUATION

**Evaluation datasets.** We evaluate on five diverse datasets spanning various scenarios: **Sintel** (Butler et al., 2012) consists of 23 synthetic videos with 50 frames each, providing precise depth labels in complex scenes with challenging lighting and motion. **ScanNet v2** (Dai et al., 2017) includes 100 indoor test videos with rich geometric structures, from which we sample every third frame to create 90-frame sequences for standard evaluation. **Bonn** (Palazzolo et al., 2019) contains 26 dynamic videos with prominent foreground motions, where we use frames 30-140 to assess robustness to object movement. **KITTI** (Geiger et al., 2013) provides 13 outdoor driving sequences, from which we use the first 110 frames per sequence from the full validation set to evaluate performance in structured environments. **DDAD** (Guizilini et al., 2020) is an autonomous driving dataset featuring diverse outdoor scenes captured across varying weather conditions and environments, with sequences ranging from 50 to 100 frames. For ablation studies focusing on long-range temporal consistency, we extend our evaluation to 270-frame sequences using consistent sampling strategies across datasets where ground truth is available.

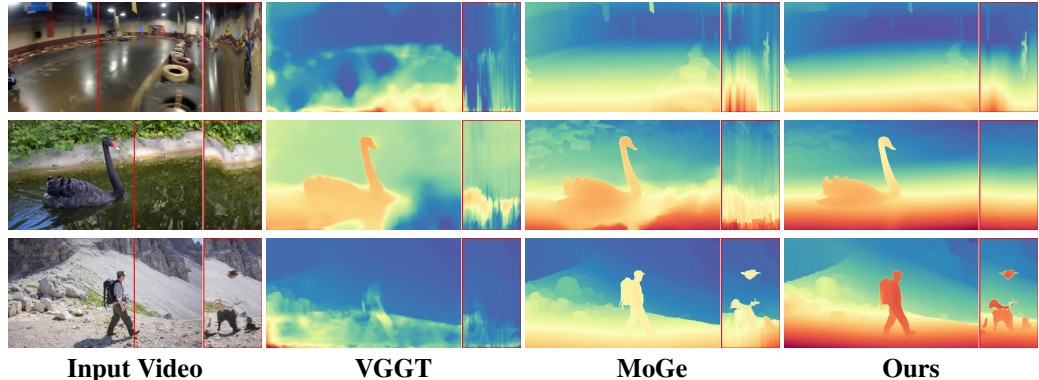

**Input Video**          **VGGT**          **MoGe**          **Ours**

Figure 3: **Qualitative visualizations of depth predictions across diverse scenarios.** Each row shows an input frame with its corresponding spacetime slice (right portion), comparing depth predictions from VGGT, MoGe, and our method.

| Inference Method | ScanNet (270 frames) | | | | KITTI (270 frames) | | | | DDAD | | | |
|---|---|---|---|---|---|---|---|---|---|---|---|---|
| | $\text{Rel}^p\downarrow$ | $\delta^p\uparrow$ | $\text{Rel}^d\downarrow$ | $\delta^d\uparrow$ | $\text{Rel}^p\downarrow$ | $\delta^p\uparrow$ | $\text{Rel}^d\downarrow$ | $\delta^d\uparrow$ | $\text{Rel}^p\downarrow$ | $\delta^p\uparrow$ | $\text{Rel}^d\downarrow$ | $\delta^d\uparrow$ |
| Sliding Window | 0.114 | 0.935 | 0.098 | 0.908 | 0.102 | 0.963 | 0.097 | 0.930 | 0.192 | 0.863 | 0.115 | 0.894 |
| Single-Pass | **0.113** | **0.937** | **0.094** | **0.913** | **0.092** | **0.974** | **0.084** | **0.963** | **0.187** | **0.879** | **0.108** | **0.916** |

Table 4: **Effectiveness of single-pass processing for long sequences.** We compare directly processing entire 270-frame sequences with our frequency-modulated position encoding (Single-Pass) against traditional sliding window approach with overlapping frames.

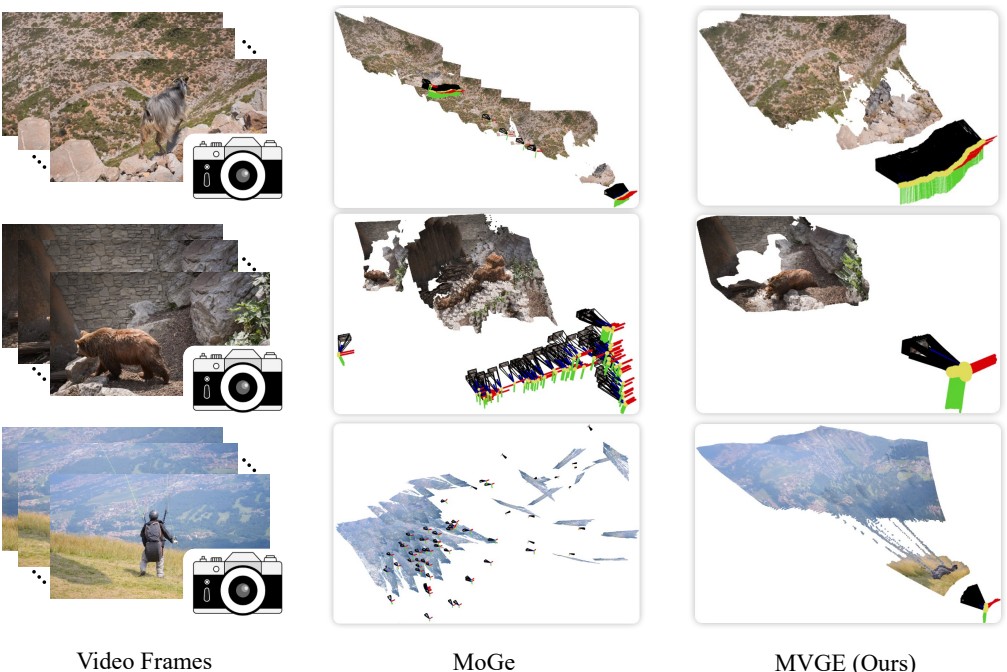

Video Frames          MoGe          MVGE (Ours)

Figure 4: **4D reconstruction comparison using MegaSAM (Li et al., 2025).** Our method enables coherent multi-view reconstruction from video sequences (right), while MoGe (middle) produces fragmented results with significant distortion. Input video frames shown on left.

**Quantitative results.** Table 1 presents our method's performance compared to state-of-the-art approaches across diverse datasets. We report both point map estimation ($\text{Rel}^p$, $\delta^p$) and depth estimation ($\text{Rel}^d$, $\delta^d$) metrics, where Rel measures relative absolute error (lower is better) and $\delta$ represents the percentage of pixels with relative error less than 1.25 (higher is better). For temporal

consistency, we employ the temporal alignment error (TAE) metric introduced by Yang et al. (2025). All evaluations use a shared scale factor for alignment across entire video sequences to fairly assess global consistency.

Our approach significantly outperforms previous methods, achieving the lowest average rank across all datasets. Specifically, we achieve substantial improvements: 8.5% $\text{Rel}^p$ reduction on Sintel, 24.2% accuracy and 34.9% temporal consistency ($\text{TAE}^p$) improvements on ScanNet, and 9.9% $\text{Rel}^p$ reduction on KITTI compared to MoGe. Depth metrics show similar trends across datasets.

Table 3 evaluates our method against state-of-the-art video depth estimation approaches. Our method achieves competitive or superior accuracy across all datasets while demonstrating remarkable computational efficiency, processing sequences 6-42× faster than existing methods.

**Qualitative comparison.** Figure 3 presents spacetime slice visualizations where our method maintains superior temporal consistency across diverse scenarios compared to VGGT and MoGe. Figure 4 demonstrates the downstream impact, with our approach enabling coherent 4D reconstructions via MegaSAM (Li et al., 2025) while MoGe produces fragmented results under identical conditions. To quantify this reconstruction quality, we evaluated camera pose accuracy using our predicted point maps as input to MegaSAM on the Sintel dataset. Our method achieves significant improvements with ATE of 0.035 and RTE of 0.014, outperforming both MonST3R (ATE: 0.078, RTE: 0.038) and MoGe (ATE: 0.087, RTE: 0.033) by 55% and 60% respectively in ATE, and 63% and 58% respectively in RTE. Notably, our method achieved 100% success rate while MoGe failed completely on 2 scenes.

### 4.3 ABLATION STUDY

**Extrapolation strategies for long sequences.** Table 2 analyzes different position encoding strategies for processing sequences significantly longer than our 24-frame training examples. We evaluate four approaches: no temporal position encoding (None), absolute position encoding (APE), standard rotary position encoding with NTK adaptation (RoPE), and our complete approach that combines NTK-adapted RoPE with sequence stretching during training to simulate extrapolation (RoPE+).

**Effectiveness of temporal and geometric constraints.** We evaluate our hierarchical temporal supervision ($\mathcal{L}_{temp}$) and cross-frame geometric constraints ($\mathcal{L}_{cross}$) on Sintel, ScanNet, and DDAD datasets. The combination of both losses achieves pointmap temporal consistency improvements of 9.53% on average across datasets and depth temporal consistency improvements of 18.4% compared to baseline MoGe constraints.

**Single-pass vs. sliding window inference.** Table 4 compares our single-pass processing approach with traditional sliding window techniques (Chen et al., 2025) for handling long sequences. Our method directly processes entire 270-frame sequences in a single forward pass. This approach not only eliminates computational redundancy but also consistently improves performance across all datasets. On KITTI, single-pass processing reduces $\text{Rel}^p$ by 9.8% compared to sliding window approaches, highlighting the benefits of maintaining global context across the entire sequence rather than processing overlapping segments independently.

**Computational efficiency.** Using an NVIDIA H20 GPU with FP16 inference, our model processes 300 frames at 378×672 resolution in 7.68 seconds (39.1 FPS) with 76.53 GB memory usage. The optimization for 300 frames uses 0.337 seconds, averaging 1.12 ms per frame.

## 5 CONCLUSION

We presented a novel approach for monocular video geometry estimation that addresses the dual challenge of high geometric accuracy and long-range temporal consistency. Our method generates scale-invariant 3D point maps through three key innovations: viewpoint-invariant geometry aligning points in a unified reference frame, appearance-invariant learning preserving structural features despite visual variations, and frequency-modulated positioning enabling extrapolation to sequences vastly exceeding training examples. Experiments demonstrate substantial improvements over state-of-the-art methods, with our efficient single-pass approach maintaining both fine-grained detail and global consistency across diverse datasets. **Limitation**: Our current approach relies on external methods (Li et al., 2025) for camera pose estimation rather than direct prediction within our model, which will be addressed in future work toward a fully end-to-end solution.

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
