# OpenReview forum: "MVGE: Scale-invariant and Temporal-consistent Monocular Video Geometry Estimation"
_ICLR.cc/2026/Conference — ICLR 2026 Conference Withdrawn Submission_

### Official Review · Reviewer_kBmV · 2025-10-31

**Soundness:** 2
**Presentation:** 2
**Contribution:** 2
**Rating:** 2
**Confidence:** 4

**Summary:**

The paper proposes a temporal-consistent monocular video depth method. Given a set of multiview frames up to hundreds of frames, the method output temporally consistent affine-invariant depth maps. With an off-the-shelf pose estimator, the method aligns the depths to pointmaps. Qualitative and quantitatively, the method outperforms state-of-the-art feedforward methods that directly output pointmaps at the canonical coordinate (eg., VGGT) or monocular depth methods (MoGe or DepthPro).

---

There are several concerns on insufficient comparison, unclear details, insufficient ablation study, and the novelty concerns. Thus the recommendation is **2: reject**. It would be appreciated if any justifications on those concerns are provided.

**Strengths:**

* **Quite interesting technical components for training**

  In section 3.1 and section 3.2 (or Figure 2), the paper introduces quite interesting technical components to extend the previous monocular depth method (MoGe) to be temporal consistent along the video frames. This include multi-scale loss, temporal consistency loss, and RoPE with dynamic NTK scaling to handle longer sequence at test time.

* **Good accuracy**

  According to Table 1, the method achieves better accuracy than direct competitors such as DepthPro, VGGT, and MoGe. The attached supplementary includes several qualitative examples. The visual quality is very good.

**Weaknesses:**

* **Insufficient comparison**

  One of the end goals of the paper is temporally consistent video depth estimation. In that regards, there are missing baselines from the video depth literatures, eg., DepthCrafter, DepthAnyVideo, Video Depth Anything, etc. Also, 3D pointmap or long-range reconstruction is the other goal. So it makes sense to compare other relevant baselines, eg., Spann3R, CUT3R, etc., or SLAM baselines, eg, DROID-SLAM, MASt3R-SLAM, DPVO, ORB-SLAM, MegaSaM.


* **Unclear details**

  The paper needs better clarity. There are lots of missing details that makes understanding technical details difficult.
  * At line 160, LHS and RHS have the same variables, $P_i,$ which maybe needs to be different
  * At line 255, what is $w_i$, depth-aware weight? There is no explanation or equation.

  Also some technical designs are not easy to understand and may need better justifications.
  * At line 255, why ($s_c$, $t_c$) are computed independently for each cell? For consistency among each cell, shouldn't they be considered together?
  * Why is multi-grid or local alignment needed in section 3.1? To be spatio-temporally consistent over all frames, isn't it enough if we have one global alignment? What benefit does the multi-grid or local alignment provide?
  * At line 259 how is the derivative ($\delta D / \delta t$) computed? Is it a difference between two depth maps at the different time steps?

* **Ablation study**

  In section 4.3, the ablation study on geometric and temporal constrains is not so clear. Would it be possible to provide it as a table with evaluation on multiple datasets? (providing other qualitative examples could be a plus). In the current presentation format (only by looking at few numbers), it is difficult to understand how strong its effectiveness is.

* **Novelty concern**

  The method uses an off-the-shelf pose estimation method, which can be an unfair advantage over other methods that jointly estimate camera pose or pointmaps at the canonical coordinate. What would be the main novelty over those methods?

**Questions:**

It's included in the weakness section above.

---

### Official Review · Reviewer_hGv8 · 2025-10-31

**Soundness:** 3
**Presentation:** 3
**Contribution:** 3
**Rating:** 2
**Confidence:** 4

**Summary:**

The paper proposes MVGE, a video model that predicts affine-invariant 3D point maps for all frames in a long monocular sequence with shared per-sequence scale/shift parameters. It combines (i) cross-frame geometric alignment using poses during training, (ii) hierarchical temporal consistency via multi-scale temporal derivative losses, and (iii) NTK-adapted RoPE with "sequence stretching" to extrapolate beyond the training context. The method reports improved point/depth accuracy and temporal alignment over MoGe, VGGT, and video-depth baselines, and demonstrates single-pass inference over hundreds of frames reasonably fast.

**Strengths:**

- A timely solution for an important problem.
- Well-motivated temporal losses across multiple strides; goes beyond adjacent-frame flow constraints.
- Single-pass inference over hundreds of frames avoiding overlapping-window redundancy.
- Good numbers in the experiments.

**Weaknesses:**

Most importantly:
- Missing SLAM comparisons. There is no quantitative comparison to modern SLAM/SGM pipelines that jointly estimate geometry and camera motion (e.g., VGGT-SLAM, Mast3R-SLAM, DPVO, DROID-SLAM, ORB-SLAM3, or recent learned SfM/SLAM hybrids). Given the focus on long sequences and drift, such baselines are crucial and the lack of them really does not allow me to judge how well the method really works.

Major things:
- Assumption of shared intrinsics (single focal) and z-shift per sequence may fail under zooming/auto-focus or rolling-shutter effects. I know many paper assume this, but this is what makes many actual SLAM method not work on long sequences of in-the-wild videos. It would be important to analyze how much the methods suffer if zooming happens.
- Memory footprint is very high (≈76 GB for 300 frames). The practical deployment constraints and scaling trade-offs (resolution vs. length vs. VRAM) are not analyses at all. If we talk about reconstructing videos, we usually mean minutes/hours long ones. This method with 30 FPS will work on 10 secs video on an A100. I am curious whether there is a way to solve the problem on smaller, more practically interesting GPUs and how would it work / degrade performance. For example, keyframing with wider frame steps, etc.

Minor things:
- Training uses ground-truth poses “optionally” for cross-frame loss, but the fraction of data using poses and its impact on generalization are unspecified. This should be provided.

**Questions:**

All in all, I like the paper but my main problem is the lack of comparison to SLAM baselines.
The authors can convince me to improve my rating by providing comparison to SOTA SLAM methods.

**Details Of Ethics Concerns:**

No concerns

---

### Official Review · Reviewer_o339 · 2025-11-01

**Soundness:** 3
**Presentation:** 3
**Contribution:** 2
**Rating:** 6
**Confidence:** 4

**Summary:**

The paper proposes MVGE, a novel framework for monocular video geometry estimation that aims to maintain both geometric accuracy and temporal consistency over long video sequences.
The authors introduce three major innovations: viewpoint-invariant geometry, appearance-invariant learning, frequency-modulated positioning. Experiments on ScanNet, KITTI, and Sintel show that MVGE improves geometric accuracy and temporal alignment by 34.9%, while running at high frequency. However, the system still depends on external pose estimation for full 3D reconstruction.

**Strengths:**

1. This paper addresses an important problem: how to maintain both geometric accuracy and temporal consistency over long video sequences.
2. The three key components of the method form a cohesive end-to-end geometry estimation pipeline.
3. Experiments show large improvements across multiple datasets, and cover diverse domains including indoor and outdoor.
4. The proposed approach is well designed for cross-frame geometric consistency and hierarchical temporal consistency.

**Weaknesses:**

1. MVGE cannot estimate camera poses directly. It relies on external methods like MegaSAM for reconstruction.
2. Theoretical grounding for NTK scaling in the pipeline is limited.

**Questions:**

1. Does the proposed pipeline account for moving objects or non-rigid motions?
2. How robust is the method when the above dynamic elements are present?
3. How does the shared scale and shift parameter assumption hold when the scene contains moving objects, depth discontinuities, or partial occlusions?
4. The temporal loss uses hierarchical derivative, does this encourage over-smoothing in rapidly changing sequences?

---

### Official Review · Reviewer_9jsx · 2025-11-06

**Soundness:** 3
**Presentation:** 4
**Contribution:** 3
**Rating:** 6
**Confidence:** 3

**Summary:**

This paper introduces MVGE, a novel method for estimating 3D geometry from monocular video sequences. The primary goal is to address the trade-off between per-frame geometric accuracy and long-range temporal consistency, a key challenge in this domain.The core contributions are threefold:
1. Viewpoint-Invariant Geometry: A cross-frame geometric consistency loss ($\mathcal{L}_{cross}$) that aligns points from different viewpoints into a common reference frame
2. Appearance-Invariant Learning: A hierarchical temporal derivative loss ($\mathcal{L}_{temp}$) that enforces consistency across exponential time scales (e.g., $\delta=1, 2, 4, 8$ frames)
3. Frequency-Modulated Positioning: A novel adaptation of Rotary Position Encoding (RoPE) with NTK-aware scaling and a "train-time sequence stretching" strategy

**Strengths:**

1. Shared sequence-level scale/shift with per-cell affine fits strikes a pragmatic balance between geometric fidelity and global consistency.
2. Standard Transformers struggle to generalize beyond trained sequence lengths. MVGE addresses this with an NTK-guided, frequency-modulated rotary positional encoding and training-time simulation over exponential time scales. This design lets its temporal attention extrapolate to sequences orders of magnitude longer than seen in training while maintaining stable attention kernels.
3. Method is decomposed into geometric, temporal, and positional-encoding components with equations and a helpful figure.

**Weaknesses:**

1. (Minor) Outputs are affine/scale-invariant; limits plug-and-play use in metric 3D tasks.
2. Long-range claim may be overstated. VGGT also handles ~300 frames in one pass as well. There are alredy works[1] that could process more than 3000 frames with the same memory constraint.
3. Asymmetric comparison. MVGE uses external poses; VGGT predicts poses, making the fairness of comparisons unclear.


[1]Deng, Kai, et al. "VGGT-Long: Chunk it, Loop it, Align it--Pushing VGGT's Limits on Kilometer-scale Long RGB Sequences." arXiv preprint arXiv:2507.16443 (2025).

**Questions:**

1. Metric scale: What prevents direct metric prediction? Would removing per-frame normalization and training with metric supervision enable it?
2. It would be nice to have an ablation of performance vs frame number.
3. Pose parity study: With identical external poses provided, how do MVGE, VGGT, and DUSt3R compare on geometry alone?
4. Temporal geometry consistency is effective in static scenes. Could the authors elaborate on how the method handles moving objects? The supplementary material also seems to demonstrate strong performance on this case.

---

### Note · Authors · 2026-01-15

I have read and agree with the venue's withdrawal policy on behalf of myself and my co-authors.